# Why Is Health Care for Children with Down Syndrome So Crucial from the First Days of Life? A Retrospective Cohort Study Emphasized Transient Abnormal Myelopoiesis (TAM) Syndrome at Three Centers

**DOI:** 10.3390/ijerph19159774

**Published:** 2022-08-08

**Authors:** Gabriela Telman, Patrycja Sosnowska-Sienkiewicz, Ewa Strauss, Jan Mazela, Przemysław Mańkowski, Danuta Januszkiewicz-Lewandowska

**Affiliations:** 1Department of Pediatric Oncology, Hematology and Transplantology, Poznan University of Medical Sciences, Szpitalna Street 27/33, 60-572 Poznań, Poland; 2Department of Pediatric Surgery, Traumatology and Urology, Poznan University of Medical Sciences, Szpitalna Street 27/33, 60-572 Poznań, Poland; 3Institute of Human Genetics, Polish Academy of Sciences, Strzeszyńska Street 32, 60-479 Poznań, Poland; 4Department of Newborns’ Infectious Diseases, Poznan University of Medical Sciences, Polna Street 33, 60-535 Poznań, Poland

**Keywords:** Down syndrome, myeloid leukemia of Down syndrome, parent education, therapeutic algorithm, transient abnormal myelopoiesis

## Abstract

Down syndrome (DS) is a common genetic disorder and is associated with an increased likelihood of many diseases, including defects of the heart, genitourinary system, gastrointestinal tract, and oncological diseases. The aim of this study was to analyze medical problems occurring in newborns with DS and to create a basic diagnostic and therapeutic algorithm intended primarily for neonatologists, pediatricians, family physicians, and physicians of other specialties caring for children with DS. Over a 5-year period, the medical records of 161 neonates with Down syndrome from four neonatology departments in Poznan, Poland, were examined. After applying exclusion criteria, 111 patients were analyzed. Data obtained from medical history included sex, week of gestation, birth weight, APGAR score, clinical symptoms, peripheral blood count with smear, and clinical features such as jaundice, hemorrhagic diathesis, ascites, hepato- or splenomegaly, pericardial or pleural effusion, respiratory failure, and other rare transient signs of abnormal myelopoiesis: fetal edema, hepatic fibrosis, renal failure, and rush. In the study group, 8% of children with Down syndrome were diagnosed with a heart and 1.8% with a genitourinary defect. Transient abnormal myelopoiesis syndrome (Transient abnormal myelopoiesis (TAM)) was found in 10% of newborns with DS. A blood count with blood smear, cardiology consultation with echocardiography, and an abdominal ultrasound should be performed in the first few days after birth in all newborns with Down syndrome. If this is not possible and the child’s condition is stable, these tests can be performed within 2–3 months after birth.

## 1. Introduction

Down syndrome (DS) is a common genetic disorder and is caused by the presence of an extra chromosome 21 or part of it [1]. Aneuploidy is most commonly caused by trisomy 21, which occurs in 95% of cases. Much less common is the mosaic variant, in which only some cells have an extra copy of chromosome 21. This variant causes 1–2% of Down syndrome cases. The phenotypic features of Down syndrome may also occur due to the so-called Robertson translocation, which is responsible for 2–3% of cases or duplication of chromosome 21 [2].

Down syndrome is associated with delay in physical growth, intellectual disability, and distinctive facial features [3,4]. The most common dysmorphic features include a flattened face, small head, short neck, protruding tongue, upward slanting eyelids, abnormally shaped or small ears, poor muscle tone, broad, short hands with a single crease on the palm, relatively short fingers and small hands and feet, and excessive flexibility in the joints [5]. Children with Down syndrome have an increased risk of congenital heart disease, hearing loss, gastrointestinal defects such as duodenal and anorectal atresia, Hirschsprung’s disease, and skeletal and genitourinary defects. They are more likely to develop eye diseases, including cataracts and severe refractive defects, hip dislocation, obstructive sleep apnea, hypothyroidism, celiac disease, epilepsy, or—later in life—Alzheimer’s disease. These children often present abnormal humoral immunity, resulting in recurrent respiratory infections and otitis media. Children with Down syndrome also have a higher risk of leukemia, so-called myeloid leukemia of Down syndrome (ML-DS) [6]. This disease is caused by a transient neonatal preleukaemic syndrome, transient abnormal myelopoiesis (TAM). TAM and ML-DS are results of cooperation between trisomy 21, which itself perturbs fetal hematopoiesis and mutations in the key hematopoietic transcription factor gene GATA1. These mutations are present in almost one-third of DS neonates and are frequently clinically and hematologically ‘silent’. TAM is characterized by an increased number of circulating blasts that contain acquired N-terminal shortening mutations in the key gene of the hematopoietic transcription factor GATA1 [7].

There is no cure for Down syndrome; therefore, proper care for these children is essential from the first days of life. Parent education and appropriate care significantly improve the quality of life of these children [8,9]. In developed countries with adequate health care, the life expectancy in people with Down syndrome is 50–60 years. Regular screening for health problems that often co-occur in Down syndrome is recommended throughout life and should begin immediately after birth. Such management is extremely important to improve the quality of life for people with Down syndrome [10].

However, to make care as useful as possible, it is necessary to develop a supportive diagnostic and therapeutic algorithm for neonatologists, obstetricians, family physicians, and physicians in other specialties. In some regions of the world, such guidelines exist [10,11]. In Poland, the diagnostic and therapeutic recommendations for children with Down syndrome are ambiguous. While congenital heart and gastrointestinal defects receive special attention in children with DS, other conditions, including hematological and immunological disorders, remain largely forgotten. The purpose of this study is to highlight the need for multidisciplinary care for children with DS shortly after birth in order to improve their quality of life. Hence, the results of the study can be used to prepare a basic diagnostic and therapeutic algorithm for any physicians who care for children with Down syndrome.

## 2. Materials and Methods

In this study, we decided to analyze the frequency of medical problems, including hematological disorders, TAM syndrome, and leukemia occurring in newborns with Down’s syndrome born in three hospital centers during the 5-year study period. It was also checked whether children with DS were referred to a hematologist or oncologist. The collected data were based on the medical records of patients from four Neonatology Departments in three medical centers in the capital of the Greater Poland Region: the Gynecology and Obstetrics University Hospital, which includes two departments: the Neonatology Department and the Neonatal Infectious Diseases Department (the newborn is hospitalized in one of them depending on the state of health), the Provincial Hospital in Poznan and the Franciszek Raszeja City Hospital. The first hospital mentioned is the largest of its kind in the region and the only one with the third degree of reference, which is the highest reachable in Poland and means that the medical center offers highly specialized procedures in convoluted cases. All hospitals treated pregnant women from both urban and rural regions. Prenatal care in Poland is offered to all pregnant women. However, the way in which it is utilized, as well as proper nutrition during pregnancy, largely depends on individual environmental conditions.

The number of total live births per year between 2016 and 2020 from these centers is listed in Table 1.

The total number of newborns with DS across all 4 hospitals was 161. From this group, after applying the exclusion criteria (another place of birth: 39 cases and multiple hospitalizations: 11 cases), a group of 111 patients was separated. The data (1) were obtained from the medical history records, (2) were collected from the first day of life, and (3) included the patients’ sex, week of gestation, birth body mass, APGAR score, health diagnoses, congenital malformations, biochemical examinations, the result of peripheral blood count with smear, and clinical features such as jaundice, bleeding diathesis, ascites, hepato- or splenomegaly, pericardial or pleural effusion, respiratory failure, and other rare TAM symptoms: fetal edema, liver fibrosis, renal failure, and rush.

Statistical analysis was performed using Statistica 10 software (StatSoft Inc., Tulsa, OK, USA). Chi2 Test, *t*-Test, and Mann–Whitney U Test were used. The observed differences were considered significant at *p* < 0.05. The data were elaborated for the group of all children, and one’s with TAM Syndrome separately.

The Bioethics Committee at the Poznan Medical University approved carrying out this study. All procedures involving human subjects were performed in accordance with 1the ethical standards of the institutional and/or national research committee and with the 1964 Declaration of Helsinki and its subsequent amendments or comparable ethical standards. We did not receive funding for this study from any public, commercial or non-profit sources.

## 3. Results

The analysis of 111 newborns with Down syndrome showed that 15 of them required further hospitalization in specialized departments immediately after birth. These were the departments of cardiology (*n* = 6), cardiac surgery (*n* = 3), oncology (*n* = 3), nephrology (*n* = 2), and pediatric surgery (*n* = 1). The duration of hospitalization of children in these departments ranged from 1 to 44 days. Two other newborns, after hospitalization in the neonatal unit, were referred to the hematology-oncology outpatient clinic for further diagnostics and observation. In summary, only 5 of 111 (4.5%) children with DS were referred for hematology-oncology care, including only two with TAM. Four children died between 10 and 29 days of life. The direct cause of death was circulatory-respiratory failure related to infection. Among all neonates with DS, jaundice, respiratory failure, and hemorrhagic diathesis were the most frequently observed abnormalities (Table 2). Among the entire study group, 11 neonates (9.9%) were diagnosed with TAM syndrome. The diagnosis of TAM was made on the basis of physical examination, peripheral blood count with smear, and biochemical tests. In none of the DS patients, despite genetic counseling, testing for GATA1 gene mutations was performed.

Hemorrhagic diathesis, hepato- and splenomegaly, pericardium and pleural effusion, respiratory and kidney failure, fetal edema, and leukemia were significantly more frequently observed in children with DS accompanied by TAM syndrome compared to children without it (Table 2). Blood count and smear results comparing newborns with Down syndrome with and without TAM showed significantly lower hemoglobin levels and platelet counts in the TAM group (Table 3). Moreover, in this group, significantly higher leukocytosis, blast and erythroblast count in peripheral blood smear were observed.

## 4. Discussion

Many congenital conditions can be present in children with DS. Although the phenotype of children with DS is variable, there are many typical features that enable physicians to make an accurate diagnosis [12]. For ethical and medical reasons, it is necessary to know about the possible diseases and problems that can occur in patients with Down syndrome from the first days of life.

One of the possible co-occurring problems in children with Down syndrome is heart defects [6]. The most common of these include ventricular septal defect, persistent ductus arteriosus, ventricular septal defect, and tetralogy of Fallot [13]. Some heart defects are diagnosed during prenatal ultrasound [6,13]. Other children with DS and major heart defects may develop heart failure, breathing difficulties, and developmental problems in the neonatal period. However, in some cases, the defect may not produce clinical symptoms; hence it is important that all children with DS have an echocardiogram at birth. If this is not possible, it should be performed within the first two or three months of life. In our group of patients with DS, all newborns had echocardiography performed. Of the 111 children with DS, six neonates required hospitalization and further diagnostics in the cardiology department immediately after birth; another three children were transferred to the cardiac surgery department. Taking into account that a total of 15 children were transferred to other specialist wards, including nine to wards with cardiology specificity, these data indicate the need for evaluation for heart defects in neonates with DS.

People with Down syndrome are highly protected against most solid tumors but have a high predisposition to certain types of blood cancers, mainly leukemia. In our study group of newborns with DS, this was the second most common condition (three newborns) requiring further hospitalization in an oncology unit and further diagnostics in an outpatient setting (two patients). Acute megakaryoblastic leukemia (AMKL), known as myeloid leukemia associated with Down’s syndrome (ML-DS), is particularly common in children with DS, and its development is closely related to the preceding so-called transient abnormal myelopoiesis (TAM) [14]. TAM, which appears to be a rare condition, can develop in approximately 4–10% of all neonates with DS. Some researchers point out that the diagnosis is underestimated due to the asymptomatic course of the disease and spontaneous remission [14]. In our study group, TAM syndrome occurred in 11 of 111 neonates, which represented 9.9% of analyzed children with DS. The GATA1 gene mutation appears to be crucial for the occurrence of TAM syndrome in neonates with and without DS [7], as well as for the development of AMKL, so its detection plays a key role. However, testing for GATA1 gene mutations is not widely used in neonatal and pediatric units in Poland. In the available records analyzed by us, none of the children were tested for GATA1 mutation.

Depending on the symptoms and clinical presentation, the prognosis in children with DS and concomitant TAM can be good; up to 80% of cases achieve spontaneous remission or inauspicious and is associated with an increased risk of death. The percentage of blasts in the peripheral blood appears to be related to the patient’s condition. Muramatsu et al. showed that the onset of life-threatening symptoms and the higher median age at which blast cells disappear from the blood are more likely in patients with a higher percentage of blast cells, whereas infants with LBP-TAM, defined as TAM syndrome with low peripheral blood blast cell with percentage ≤10%, tended to be in better condition and were less likely to develop AMKL or premature death [15]. Therefore, it is important to perform a complete blood count with peripheral blood smear in all neonates with DS. These tests should be repeated multiple times because up to 30% of patients with TAM syndrome may develop AMKL. Prevention carried out in this way can avoid delaying the diagnosis and treatment of leukemia [7].

Children with Down syndrome often have genitourinary malformations [16,17]. In our study, this was the third most common condition (two newborns) requiring further hospitalization in the nephrology department. Kupferman et al., in their study, compared 3832 children with Down syndrome and 3,411,833 healthy children. The study showed that 3.2% of children with DS had kidney and urinary tract anomalies compared to 0.7% of children without DS. The defects observed in children with Down syndrome included anterior urethral obstruction, renal cystadenopathy, hydronephrosis, ureters, hypospadias, posterior urethral valves, prune syndrome, and renal agenesis [18]. The children in our study had posterior urethral valves and prune syndrome. They accounted for 1.8% of newborns with DS in the entire study group.

According to the extant literature, approximately 3% of infants with Down syndrome are born with an imperforate anus, and about 5% of infants with DS are born with Hirschsprung disease [19]. One neonate in our group failed to pass meconium within 48 h. Due to abdominal enlargement, vomiting, and no clinical improvement after conservative treatment, he was referred to the pediatric surgery department. The child underwent bowel mapping, and a stoma was placed at the splenic flexure. Histopathological examination confirmed Hirschsprung’s disease.

The available literature data do not allow sufficient comparison of differences in demographic characteristics of population samples. In the last 10 years, no data on this subject have been published from Central Europe, and with regard to Eastern Europe, no such data have been found at all. Poland, together with the Czech Republic, Slovakia, and Hungary, belongs to the Visegrad Group countries, of which the presented research on the subject is the first. Nevertheless, the findings presented to correspond with the cited publications and confirm the diagnosis of TAM in 10% of infants with DS.

Careful prenatal care and appropriate screening can lead to a highly probable suspicion of DS in the fetus. However, estimating the screening rate for prenatal DS confirmation was not an expected result; after collected data analysis, it occurred that only in less than 10% of infants DS was prenatally diagnosed. This may be due to false-negative ultrasound results, failure to perform amniocentesis, or prenatal genetic testing. Poland’s National Health System guarantees regular gynecological and obstetric care during pregnancy, but underfunding of medical procedures is clearly noticeable. It is highly important to emphasize the need for public health awareness about insightful prenatal screening for DS so that both parents and clinicians could be well prepared for possible health challenges to come in their children or patients.

## 5. Conclusions

In conclusion, newborns with Down syndrome require exceptional care supported by adequate medical expertise. On the basis of our analysis, we believe that the following examinations are mandatory in the first days of life: blood count and peripheral blood smear, cardiological consultation and cardiac echocardiography, and abdominal ultrasound. If the above-mentioned examinations and consultations cannot be performed after birth, doctors and parents should make sure to perform them within 2–3 months after birth. Proper diagnosis and care of a child with Down syndrome will improve their living conditions and chances for the best possible development.

## Figures and Tables

**Table 1 ijerph-19-09774-t001:** Number of all live births and newborns with DS (in parentheses) between 2016 and 2020 in Gynecological and Obstetrics University Hospital (UH), Provincial Hospital of Poznan (PH), and Franciszek Raszeja City Hospital (CH).

Year	Number of Life Births (Newborns with DS)			
	UH	PH	CH	Total Per Year
2016	7824 (22)	1420 (1)	2532 (1)	11,776 (24)
2017	7742 (22)	1542 (1)	2652 (2)	11,936 (25)
2018	7052 (17)	1585 (2)	2597 (1)	11,234 (20)
2019	6934 (18)	1626 (1)	2520 (1)	11,080 (20)
2020	6583 (19)	1545 (1)	2183 (2)	10,311 (22)

**Table 2 ijerph-19-09774-t002:** The comparison of symptoms occurrence that might lead to suspicion of TAM.

Analyzed Parameter, *n* (%)	All Newborns with DS (*n* = 111)	DS no TAM (*n* = 100)	DS with TAM (*n* = 11)	*p*-Value
Birth weight (g)				
Mean (SD)	2812.3 (712.8)	2814.3 (728.2)	2794.2 (582.4)	0.930
Min–Max	765–4450	765–4450	1900–3590	
Median (Q25; Q75)	2950 (2260; 3320)	2955 (2295; 3320)	2880 (2256; 3360)	0.836
Jaundice	87 (78.4)	78 (78.0)	9 (81.8)	0.770
Hemorrhagic diathesis	51 (45.9)	41 (41.0)	10 (90.9)	0.002
Ascites	2 (1.8)	2 (2.0)	0 (0.0)	0.636
Hepatomegaly	7 (6.3)	3 (3.0)	4 (36.4)	<0.001
Splenomegaly	2 (1.8)	0 (0.0)	2 (18.2)	<0.001
Pericardium effusion	2 (1.8)	0 (0.0)	2 (18.2)	<0.001
Pleural effusion	2 (1.8)	1 (1.0)	1 (9.1)	0.056
Respiratory failure	65 (58.6)	54 (54.0)	11 (100.0)	0.003
Fetal edema	3 (2.7)	1 (1.0)	2 (18.2)	<0.001
Kidney failure	5 (4.5)	3 (3.0)	2 (18.2)	0.021
Death	5 (4.5)	3 (3.0)	2 (18.2)	0.021
Leukemia	9 (8.1)	3 (3.0)	6 (54.6)	<0.001

DS: Down syndrome, TAM: transient abnormal myelopoiesis syndrome.

**Table 3 ijerph-19-09774-t003:** Peripheral blood count and peripheral blood smear results in newborns with Down syndrome with and without TAM.

Analyzed Parameter	All Newborns with DS (*n* = 111)	DS no TAM (*n* = 100)	DS with TAM (*n* = 11)	*p*-Value
HGB (g/L)				
Mean (SD)	13.19 (1.72)	13.3 (1.5)	12.1 (3.0)	0.021
Min–Max	5.2–18.5	8.3–18.5	5.2–14.5	
Median (Q25; Q75)	13.3 (12.4; 14.3)	13.3 (12.5; 14.4)	13.4 (10.9; 14.1)	0.359
PLT (thousand/μL)				
Mean (SD)	154.3 (74.0)	158.1 (70.4)	119.9 (98.8)	0.105
Min–Max	25.0–390.0	33.0–390.0	25.0–311.0	
Median (Q25; Q75)	149 (108.0; 198.0)	151.5 (112.5; 201.0)	79.0 (51.0; 177.0)	0.045
WBC (thousand/μL)				
Mean (SD)	21.14 (13.4)	19.2 (7.6)	38.5 (32.1)	<0.001
Min–Max	4.95–99.99	5.0–50.0	8.5–100.0	
Median (Q25; Q75)	18.9 (14.8; 23.8)	18.6 (14.7; 23.2)	27.6 (19.82; 43.44)	0.013
Erythroblasts (×/100WBC)				
Mean (SD)	39.72 (72.5)	33.7 (62.7)	91.6 (122.7)	0.016
Min–Max	1.0–504.0	1.0–504.0	21.0–422.0	
Median (Q25; Q75)	20.5 (9.5; 41.5)	15.0 (8.0; 39.0)	42.0 (28.0; 94.0)	0.003
Blasts (%) Median (Q25; Q75) *	13.5 (9.0; 32.0)	12.0 (7.0; 27.0)	54.0 (43.0; 65.0)	0.013

DS: Down syndrome, TAM: transient abnormal myelopoiesis syndrome, WBC: white blood cells, PLT: platelets, HGB: hemoglobin; * median analysis due to data availability.

## Data Availability

Not applicable.

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
