# Peer review of "Why Is Health Care for Children with Down Syndrome So Crucial from the First Days of Life? A Retrospective Cohort Study Emphasized Transient Abnormal Myelopoiesis (TAM) Syndrome at Three Centers"

_ijerph, 2022, doi:10.3390/ijerph19159774_

Round 1
Reviewer 1 Report
This study highlights the complex and heterogenous medical comorbidities associated with Down syndrome. The study covers a wide range of medical conditions that warrant identification at or near birth and follow-up testing or intervention.
To emphasize the rarity and gravity of these medical issues, the authors should provide prevalence data on newborns without DS. Only examining health differences between DS newborns with and without transient abnormal myelopoiesis (TAM) in Table 2 gives the reader the impression that only newborns with DS and TAM are at risk for developing more serious conditions or need interventions.
My other suggestions are primarily regarding wording.
1. Abstract: DS is a genetic disorder, not a developmental disorder.
2. Introduction, Line 70 - it is better to say regular screening is important for quality of life, than "for a comfortable life".
3. Introduction, Lines 82-84 are too similar to lines 72-73. Perhaps just use the term "physicians" rather than re-listing all the different specialists.
4. Material and Methods:
The authors should describe demographic information about their sample. Are the medical centers located in urban or rural areas. Do most mothers receive prenatal care and have good nutrition during pregnancy? Readers will want to know how generalizable their findings are to newborns with DS in other countries.
"Data" is preferable over "research material" in line 86.
Authors state records were obtained from 4 neonatal wards of 3 medical centers but only list the 3 medical centers. Explain why and which center had two neonatal wards.
Second sentence should specify the number of total live births PER YEAR between 2016 and 2020 is LISTED in Table 1.
Line 97 should be "multiple hospitalization" not "multiplied".
APGAR "score" is the convention used in the U.S., not APGAR scale.
Line 99, diagnosis should be clarified as "health diagnoses"
No need to mention that Microsoft Excel was used. The statistical program that was used is the one that is relevant.
5. Results:
Use "outpatient clinic" instead of "out-patients clinic".
Line 127, list percentage of neonates diagnosed with TAM.
Tables 2 and 3, the p-values should be consistently at the hundredths or thousandths place after the decimal.
6. Discussion:
The statement about the phenotype of children with DS being variable is confusing. Do the authors mean that children with DS have variable health conditions? In the introduction, the authors described the common dysmorphic features of DS, suggesting that there's a known phenotype.
Second paragraph should specify "co-occurring problems" in children with Down syndrome, "problems" is too broad.
Line 151 seems unnecessary in the context of the other information in the paragraph.
Since this is for an international audience, the authors should spend some time explaining some of the cultural or psychosocial differences and similarities between their sample and other European countries, as well as about the healthcare system in Poland. How generalizable are the findings?
Also, the authors cite prevalence and risk rates for various conditions based upon the extant literature. They should mention how closely the samples from those studies matched theirs. If there isn't much data from countries or cities that have similar demographic characteristics as theirs, that's fine. All the more reason to highlight the paucity of information regarding newborns or children with DS in eastern European countries.
Author Response
Dear Reviewer,
We are very grateful for your review of our article “Why is health care for children with Down syndrome so crucial from the first days of life? A retrospective cohort study emphasized transient abnormal myelopoiesis (TAM) syndrome at three centers. We would like to respond to the comments and suggestions made.
This study highlights the complex and heterogenous medical comorbidities associated with Down syndrome. The study covers a wide range of medical conditions that warrant identification at or near birth and follow-up testing or intervention. To emphasize the rarity and gravity of these medical issues, the authors should provide prevalence data on newborns without DS. Only examining health differences between DS newborns with and without transient abnormal myelopoiesis (TAM) in Table 2 gives the reader the impression that only newborns with DS and TAM are at risk for developing more serious conditions or need interventions.
We agree that newborns with DS without TAM are also at risk of developing more serious conditions or require intervention, but congenital heart or gastrointestinal diseases, for example, usually require urgent medical intervention, as shown in the results section. The purpose of the paper was to focus on the health care of children with DS and to draw attention to the need for widespread awareness of it. An in-depth analysis of the medical records of newborns with DS has resulted in the alarming conclusion that most parents of children with DS are not even aware of the need for comprehensive and multidisciplinary care.
My other suggestions are primarily regarding wording.
- Abstract: DS is a genetic disorder, not a developmental disorder.
- Introduction, Line 70 - it is better to say regular screening is important for quality of life, than "for a comfortable life".
- Introduction, Lines 82-84 are too similar to lines 72-73. Perhaps just use the term "physicians" rather than re-listing all the different specialists.
4."Data" is preferable over "research material" in line 86.
- Authors state records were obtained from 4 neonatal wards of 3 medical centers but only list the 3 medical centers. Explain why and which center had two neonatal wards.
- Second sentence should specify the number of total live births PER YEAR between 2016 and 2020 is LISTED in Table 1.
- Line 97 should be "multiple hospitalization" not "multiplied".
- APGAR "score" is the convention used in the U.S., not APGAR scale.
- Line 99, diagnosis should be clarified as "health diagnoses"
- No need to mention that Microsoft Excel was used. The statistical program that was used is the one that is relevant.
- Results:
Use "outpatient clinic" instead of "out-patients clinic".
Line 127, list percentage of neonates diagnosed with TAM.
All of the aforementioned comments noted by the reviewer are included and corrected in the manuscript and explained later in my response. All corrections have been marked in yellow in the text of the paper. Thank you very much for the opportunity to improve the language of the manuscript.
The authors should describe demographic information about their sample. Are the medical centers located in urban or rural areas. Do most mothers receive prenatal care and have good nutrition during pregnancy? Readers will want to know how generalizable their findings are to newborns with DS in other countries. (…) Authors state records were obtained from 4 neonatal wards of 3 medical centers but only list the 3 medical centers. Explain why and which center had two neonatal wards.
Thank you for your suggestion and guiding questions. The revised paragraph of the Materials and Methods section reads as follows:
The collected data was based on the medical records of patients from four Neonatology Departments in three medical centers in the capital of Wielkopolska: the Gynecology and Obstetrics University Hospital - which includes two departments: the Neonatology Department and the Neonatal Infectious Diseases Department (the newborn is hospitalized in one of them depending on the state of health), the Provincial Hospital in Poznań and the Franciszek Raszei City Hospital. The first hospital mentioned is the largest of its kind in the region and the only one with the third degree of reference. All hospitals treated pregnant women from both urban and rural regions. Prenatal care in Poland is offered to all pregnant women. However, the way in which it is utilized, as well as proper nutrition during pregnancy, largely depends on individual environmental conditions.
The statement about the phenotype of children with DS being variable is confusing. Do the authors mean that children with DS have variable health conditions? In the introduction, the authors described the common dysmorphic features of DS, suggesting that there's a known phenotype.
Thank you very much for your comments. We agree, that for children with DS there are known common phenotypic features. However, problems involving, for example, cardiac, gastrointestinal, endocrine or hematological defects occur with varying frequency, so the use of the term "variable" seems appropriate.
The sentence was also corrected to: One of the possible co-occuring problems in children with Down syndrome is heart defects.
Tables 2 and 3, the p-values should be consistently at the hundredths or thousandths place after the decimal
Thank you for this comment. We have standardized the P values in Tables 2 and 3 to the thousandth decimal place. Corrected sections in the tables have been marked in yellow.
Since this is for an international audience, the authors should spend some time explaining some of the cultural or psychosocial differences and similarities between their sample and other European countries, as well as about the healthcare system in Poland. How generalizable are the findings? Also, the authors cite prevalence and risk rates for various conditions based upon the extant literature. They should mention how closely the samples from those studies matched theirs. If there isn't much data from countries or cities that have similar demographic characteristics as theirs, that's fine. All the more reason to highlight the paucity of information regarding newborns or children with DS in eastern European countries.
Thank you very much for this comment. We can clearly see the need to clarify more details for the public. Two paragraphs were added to the discussion:
The available literature data does not allow sufficient comparison of differences in demographic characteristics of population samples. In the last 10 years, no data on this subject has been published from Central Europe, and with regard to Eastern Europe, no such data has been found at all. Poland, together with the Czech Republic and Slovakia and Hungary, belongs to the Visegrad Group countries, of which the presented research on the subject is the first. Nevertheless, the findings presented correspond with the cited publications and confirm the diagnosis of TAM in 10% of infants with DS.
Careful prenatal care and appropriate screening can lead to a highly probable suspicion of DS in the fetus. The data collected confirmed that only in less than 10% of infants DS was prenatally diagnosed. This may be due to false-negative ultrasound results, failure to perform amniocentesis or genetic prenatal testing. Poland's National Health System guarantees regular gynecological and obstetric care during pregnancy but underfunding of medical procedures is clearly noticeable.
Once again, thank you for your review and valuable comments.
Yours faithfully,
Patrycja Sosnowska-Sienkiewicz, MD, PhD
Danuta Januszkiewicz-Lewandowska, MD, PhD

Reviewer 2 Report
Introduction
Introduction is very informative and concise.
In line 44 is sentence “Down syndrome is associated with physical growth retardation,..“: The term „retardation“ has negative connotations. So, please give the another term.
The paragraph in line 77 –80 „Therefore, in this study we decided to analyze the frequency of medical problems, including hematological disorders, TAM syndrome and leukemia occurring in newborns with Down's syndrome born in three hospital centers during the 5 years. It was also checked whether children with DS were referred to a hematologist or oncologist.“ should be in Methods, while in introduction should shortly describe the reason for this study.
Also, at the end of introduction it is recommended to write the aim of this study.
Materials and Methods
In Table 1. it is very useful to show the data Total number of births and also data about the number of births of children with DS, in three Centers with total number presentations.
The authors write “Of all hospitalized newborns the ones with DS were selected, giving a total number 95 of 161 children”. Which were the inclusion criteria for all children born with DS from all three centers? Please describe.
Also, it is not clear if the author included children immediately after birth or later. What are the ages of children with DS, what is the average age (or day) of children (newborn) with DS with TAM?
Results
Table 2 is without a title of the table!?!
Also, why authors used p values with 5 decimals, for example 0,00002?
Also, for data in tables the authors used comma (,) instead full stop (.)
For example, the authors write in table 3. 154,3 (74,0).
In English language number data need to be write: 154.3 (74.0)
Discussion
The author should better describe the association own results with other similar studies. This is very important for discussion.
Ethical consideration
The authors did not seek informed consent or ethical committee approval for their study. They write: „Due to the retrospective and non-invasive nature of the study, the consent of the Bioethics Committee at the Poznan Medical University was not required.“ Why do the authors not have ethical approval?? The reason which they noticed may be reasonable but, I think that is necessary because it is a very sensitive research population and very sensitive data. However, is not needed informed consent of all patient (parents) because it is retrospective and it is difficult to get, but ethical approval from Ethical committee is needed. Hence, authors must confirm they have obtained ethical approval from an appropriate ethics review board to conduct the study.
This reason is crucial for publication in this journal!
If authors do not have the ethical approval it is not seem suitable for publication in this journal.
Author Response
Dear Reviewer,
We are very grateful for your review of our article “Why is health care for children with Down syndrome so crucial from the first days of life? A retrospective cohort study emphasized transient abnormal myelopoiesis (TAM) syndrome at three centers. We would like to respond to the comments and suggestions made.
Introduction
Introduction is very informative and concise.
In line 44 is sentence “Down syndrome is associated with physical growth retardation,..“: The term „retardation“ has negative connotations. So, please give the another term.
Thank you for your valuable comments. We have changed the sentence to: Down syndrome is associated with delay in physical growth.
The paragraph in line 77 –80 „Therefore, in this study we decided to analyze the frequency of medical problems, including hematological disorders, TAM syndrome and leukemia occurring in newborns with Down's syndrome born in three hospital centers during the 5 years. It was also checked whether children with DS were referred to a hematologist or oncologist.“ should be in Methods, while in introduction should shortly describe the reason for this study.
Also, at the end of introduction it is recommended to write the aim of this study.
Thank you for your valuable comments We move the paragraph to the methods. The aim in the end of Introduction was added:
The purpose of this study is to highlight the need for multidisciplinary care for children with DS shortly after birth in order to improve their quality of life. Hence, the results of the study can be used to prepare a basic diagnostic and therapeutic algorithm for any physicians who care for children with Down syndrome.
Materials and Methods
In Table 1. it is very useful to show the data Total number of births and also data about the number of births of children with DS, in three Centers with total number presentations.
Thank you for suggestion to complete the first table. After adding missing data, the table presents like below:
|
Year |
Number of life births (newborns with DS) |
|||
|
UH |
PH |
CH |
Total per year |
|
|
2016 |
7824 (22) |
1420 (1) |
2532 (1) |
11776 (24) |
|
2017 |
7742 (22) |
1542 (1) |
2652 (2) |
11936 (25) |
|
2018 |
7052 (17) |
1585 (2) |
2597 (1) |
11234 (20) |
|
2019 |
6934 (18) |
1626 (1) |
2520 (1) |
11080 (20) |
|
2020 |
6583 (19) |
1545 (1) |
2183 (2) |
10311 (22) |
The authors write “Of all hospitalized newborns the ones with DS were selected, giving a total number 95 of 161 children”. Which were the inclusion criteria for all children born with DS from all three centers? Please describe. Also, it is not clear if the author included children immediately after birth or later. What are the ages of children with DS, what is the average age (or day) of children (newborn) with DS with TAM?
Thank you for your valuable comment. The inclusion criteria were: diagnosis of DS and one listed hospitals as a place of birth. The data were collected just after childbirth. After some modifications andr adding missing data, the text presents like below:
Of all hospitalized infants who were born in one of the above-mentioned medical centers, the newborns with DS were selected, giving a total number of 161 children. From this group, after applying the exclusion criteria (another place of birth: 39 cases and multiple hospitalization: 11 cases), a group of 111 patients was separated. The data 1) was obtained from the medical history records, 2) was collected from the first day of life and 3) included the patients’ sex, week of gestation, birth body mass, APGAR score, health diagnoses, congenital malformations, biochemical examinations, the result of peripheral blood count with smear, and clinical features like jaundice, bleeding diathesis, ascites, hepato- or splenomegaly, pericardial or pleural effusion, respiratory failure, and other rare TAM symptoms: fetal edema, liver fibrosis, renal failure, and rush.
Results
Table 2 is without a title of the table!?!
Also, why authors used p values with 5 decimals, for example 0,00002?
Also, for data in tables the authors used comma (,) instead full stop (.)
For example, the authors write in table 3. 154,3 (74,0).
In English language number data need to be write: 154.3 (74.0)
Thank you for these very important and crucial notes. I would like to ensure you that all comas in tables were changed into full stop. We have standardized the P values in Tables 2 and 3 to the thousandth decimal place. Corrected sections in the tables have been marked in yellow
The title of second table is added:
Table 2. The comparison of symptoms occurrence which might lead to suspicion of TAM.
Discussion
The author should better describe the association own results with other similar studies. This is very important for discussion.
Thank you very much for this comment. We can clearly see the need to clarify more details for the public. Two paragraphs were added to the discussion:
The available literature data does not allow sufficient comparison of differences in demographic characteristics of population samples. In the last 10 years, no data on this subject has been published from Central Europe, and with regard to Eastern Europe, no such data has been found at all. Poland, together with the Czech Republic and Slovakia and Hungary, belongs to the Visegrad Group countries, of which the presented research on the subject is the first. Nevertheless, the findings presented correspond with the cited publications and confirm the diagnosis of TAM in 10% of infants with DS.
Careful prenatal care and appropriate screening can lead to a highly probable suspicion of DS in the fetus. The data collected confirmed that only in less than 10% of infants DS was prenatally diagnosed. This may be due to false-negative ultrasound results, failure to perform amniocentesis or genetic prenatal testing. Poland's National Health System guarantees regular gynecological and obstetric care during pregnancy but underfunding of medical procedures is clearly noticeable.
Ethical consideration
The authors did not seek informed consent or ethical committee approval for their study. They write: „Due to the retrospective and non-invasive nature of the study, the consent of the Bioethics Committee at the Poznan Medical University was not required.“ Why do the authors not have ethical approval?? The reason which they noticed may be reasonable but, I think that is necessary because it is a very sensitive research population and very sensitive data. However, is not needed informed consent of all patient (parents) because it is retrospective and it is difficult to get, but ethical approval from Ethical committee is needed. Hence, authors must confirm they have obtained ethical approval from an appropriate ethics review board to conduct the study. This reason is crucial for publication in this journal! If authors do not have the ethical approval it is not seem suitable for publication in this journal.
Thank you for this valuable attention. In fact, we got the opinion of the Bioethics Committee of the Medical University of Poznań that our research is not a medical experiment. I would like to share a scan of this document below. Hence, we changed our mind in the method paragraph to: The Bioethics Committee at the Poznan Medical University has agreed to carry out this study.
Once again, thank you for your review and valuable comments.
Yours faithfully,
Patrycja Sosnowska-Sienkiewicz, MD, PhD
Danuta Januszkiewicz-Lewandowska, MD, PhD

Reviewer 3 Report
This study can be very useful for people who are interested in this topic, and also for the people in the relevant field as well, since the findings of this study provide very significant suggestions for newborns with Down syndrome. In addition, this study put a great emphasis on the importance of proper diagnosis and care of a child with Down syndrome for their better living conditions. Thus, I believe that this study can be beneficial to the relevant field, and also it perfectly fits to the aim and scope of this journal for sure.
Author Response
Dear Reviewer,
We are very grateful for your review of our article “Why is health care for children with Down syndrome so crucial from the first days of life? A retrospective cohort study emphasized transient abnormal myelopoiesis (TAM) syndrome at three centers Thank you for your friendly review.
This study can be very useful for people who are interested in this topic, and also for the people in the relevant field as well, since the findings of this study provide very significant suggestions for newborns with Down syndrome. In addition, this study put a great emphasis on the importance of proper diagnosis and care of a child with Down syndrome for their better living conditions. Thus, I believe that this study can be beneficial to the relevant field, and also it perfectly fits to the aim and scope of this journal for sure.
Once again, thank you for your review and valuable comments.
Yours faithfully,
Patrycja Sosnowska-Sienkiewicz, MD, PhD
Danuta Januszkiewicz-Lewandowska, MD, PhD

Reviewer 4 Report
The paper takes up an important research problem. The purpose of the article was defined as “analyze medical problems occurring in newborns with DS and to create a basic diagnostic and therapeutic algorithm, intended primarily for neonatologists, pediatricians, family physicians and physicians of other specialties caring for children with DS”. The assumed goal was achieved. The research results broaden the knowledge in the discussed area. It would be worthwhile to make greater use of the latest subject literature. It is also worth expanding the introduction and conclusions.
Author Response
Dear Reviewer,
We are very grateful for your review of our article “Why is health care for children with Down syndrome so crucial from the first days of life? A retrospective cohort study emphasized transient abnormal myelopoiesis (TAM) syndrome at three centers Thank you for your friendly review.
The paper takes up an important research problem. The purpose of the article was defined as “analyze medical problems occurring in newborns with DS and to create a basic diagnostic and therapeutic algorithm, intended primarily for neonatologists, pediatricians, family physicians and physicians of other specialties caring for children with DS”. The assumed goal was achieved. The research results broaden the knowledge in the discussed area. It would be worthwhile to make greater use of the latest subject literature. It is also worth expanding the introduction and conclusions.
We have made changes and additional paragraphs in the manuscript (highlighted in yellow), which is attached.
Once again, thank you for your review and valuable comments.
Yours faithfully,
Patrycja Sosnowska-Sienkiewicz, MD, PhD
Danuta Januszkiewicz-Lewandowska, MD, PhD

Round 2
Reviewer 1 Report
The overall content and aims are clearly presented. There are a few areas that need further clarification and some wording changes that I recommend.
Abstract: the word hospitalized seems redundant. The sentence is incomplete. It should state something like the medical records of 161 neonates with Down syndrome from four neonatology departments were examined. Also, the term TAM needs to be fully written out the first time it is presented.
Introduction: Second paragraph - since you are describing the conditions DS children are likely to develop, insert a clause before Alzheimer's disease (e.g., ...celiac disease, epilepsy, and later in life, Alzheimer's disease).
Third paragraph - Change to "Regular screening for health problems that often co-occur in Down syndrome..." and "Such management is extremely important to improve the quality of life..."
Fourth paragraph - change double negative sentence "there are no unambiguous diagnostic and therapeutic recommendations" to "the diagnostic and therapeutic recommendations are ambiguous". Also revise to ...newborns born in three hospital centers during the "5-year study period".
Materials and Methods: the phrase third degree of reference is unclear. It's not a common term used in English. Please explain.
The first sentence after Table 1 should be more concise -- e.g., the total number of newborns with DS across all 4 hospitals was 161.
Last paragraph should be the Bioethics Committee...approved this study. Delete "performed in this study" from the second sentenced.
Discussion: First paragraph - Instead of "For this to be possible", I suggest something like "For ethical and medical reasons".
Second paragraph middle sentence is confusing. Are you saying that children with other types of heart defects are likely to develop heart failure, etc.? The sentence needs to be reworded. Last sentence in that paragraph, use "evaluation" rather than "analysis" [for heart defects].
In third paragraph, last sentence, better to say the records that were available or records that were reviewed rather than "material".
Fourth paragraph, it is not clear what is meant by "inauspicious when associated with an increased risk of death".
Fifth paragraph, I think it would be better to call it the third most common condition rather than the third most common group.
Sixth paragraph, it is unclear at first if you are referring to the extant literature or to the medical records in your study.
Last paragraph, I recommend rephrasing sentence to "Less than 10% of the infants with DS were diagnosed prenatally". You didn't offer a hypothesis about the screening rate prenatally in your introduction so the data didn't "confirm" a prediction. I also recommend you end that paragraph emphasizing the need for public health awareness about the importance of prenatal screening for DS so parents and clinicians will request it and be informed about the possible health challenges to come.
Author Response
Dear Reviewer,
We are very grateful for your reevaluation of our article “Why is health care for children with Down syndrome so crucial from the first days of life? A retrospective cohort study emphasized transient abnormal myelopoiesis (TAM) syndrome at three centers. We would like to respond to the comments and suggestions made.
Abstract: the word hospitalized seems redundant. The sentence is incomplete. It should state something like the medical records of 161 neonates with Down syndrome from four neonatology departments were examined. Also, the term TAM needs to be fully written out the first time it is presented.
Thank you for these suggestions, they were conscientiously realized.
Introduction: Second paragraph - since you are describing the conditions DS children are likely to develop, insert a clause before Alzheimer's disease (e.g., ...celiac disease, epilepsy, and later in life, Alzheimer's disease).
Thank you for pointing this – suggested clause was added.
Third paragraph - Change to "Regular screening for health problems that often co-occur in Down syndrome..." and "Such management is extremely important to improve the quality of life...".
Thank you, the third paragraph was updated with your recommendations.
Fourth paragraph - change double negative sentence "there are no unambiguous diagnostic and therapeutic recommendations" to "the diagnostic and therapeutic recommendations are ambiguous". Also revise to ...newborns born in three hospital centers during the "5-year study period".
Thank you for these suggestions, the changes were made. Also – reffering to another Reviewer, two sentences (”In this study we decided to analyze the frequency of medical problems, including hematological disorders, TAM syndrome and leukemia occurring in newborns with Down's syndrome born in three hospital centers during the 5-year study period. It was also checked whether children with DS were referred to a hematologist or oncologist.”) were moved to Materials and Methods section.
Materials and Methods: the phrase third degree of reference is unclear. It's not a common term used in English. Please explain.
Thank you for pointing this unclearness. The explanation was made:
”The first hospital mentioned is the largest of its kind in the region and the only one with the third degree of reference, which is the highest reachable in Poland and means, that the medical center offers highly specialized procedures in convoluted cases.”
The first sentence after Table 1 should be more concise -- e.g., the total number of newborns with DS across all 4 hospitals was 161.
Thank you for this clue, your suggestion was used.
Last paragraph should be the Bioethics Committee...approved this study. Delete "performed in this study" from the second sentenced.
Thank you for these recommendations, the changes were made.
Discussion: First paragraph - Instead of "For this to be possible", I suggest something like "For ethical and medical reasons".
Thank you for this suggestion, the text was edited.
Second paragraph middle sentence is confusing. Are you saying that children with other types of heart defects are likely to develop heart failure, etc.? The sentence needs to be reworded. Last sentence in that paragraph, use "evaluation" rather than "analysis" [for heart defects].
Thank you for this question, the text could lead to kind of misunderstanding – developing heart failure depends of many conditions and some may be diagnosed by echocardiogram (if so – further medical procedures may be necessary) and that was the point of these sentences. A few words were added hoping make the text more clear:
”Some heart defects are diagnosed during prenatal ultrasound [6, 13]. In other children with DS and major heart defects may develop heart failure, breathing difficulties, and developmental problems in the neonatal period. However, in others in some cases, the defect may not produce clinical symptoms, hence it is important that all children with DS have an echocardiogram at birth.”
The recommended change in last sentence was made.
In third paragraph, last sentence, better to say the records that were available or records that were reviewed rather than "material".
Thank you for this suggestion, the change was made.
Fourth paragraph, it is not clear what is meant by "inauspicious when associated with an increased risk of death".
Thank you for pointing this unclearness. The words ”and is” are now used instead of ”when”.
Fifth paragraph, I think it would be better to call it the third most common condition rather than the third most common group.
Thank you for suggestiong this rewording, it was used in the text.
Sixth paragraph, it is unclear at first if you are referring to the extant literature or to the medical records in your study.
Thank you for highlighting that unclearness, just at the beginning the phrase ”According to extant literature …” was added.
Last paragraph, I recommend rephrasing sentence to "Less than 10% of the infants with DS were diagnosed prenatally". You didn't offer a hypothesis about the screening rate prenatally in your introduction so the data didn't "confirm" a prediction. I also recommend you end that paragraph emphasizing the need for public health awareness about the importance of prenatal screening for DS so parents and clinicians will request it and be informed about the possible health challenges to come.
Thank you for these pieces of advice, the further explanation was made and an additional, suggested sentence was added in the end of paragraph:
”However estimating the screening rate for prenatal DS confirmation was not an expected result, after collected data analysis it occurred, The data collected confirmed that only in less than 10% of infants DS was prenatally diagnosed. (…) It is highly important to emphasize the need of public health awareness about insightful prenatal screening for DS, so that both parents and clinicians could be well prepared for possible health challenges to come in their children or patients.”
Once again, thank you very much for your re-evaluation and such valuable comments.
With best regards,
Danuta Januszkiew

Reviewer 2 Report
I have completed my reevaluation of a manuscript titled „Why is health care for children with Down syndrome so crucial from the first days of life? A retrospective cohort study emphasized transient abnormal myelopoiesis (TAM) syndrome at three centers“
The revised version of the current manuscript is a significant improvement from the original. I thank the authors for their careful consideration to each comment. I have only some minor revisions that I believe still require attention. I outline these below.
Introduction
The authors in your response noticed that they moved the paragraph „Therefore, in this study we decided to analyze the frequency of medical problems, including hematological disorders, TAM syndrome and leukemia occurring in newborns with Down's syndrome born in three hospital centers during the 5 years. It was also checked whether children with DS were referred to a hematologist or oncologist.“ to the methods but this was not deleted and not moved. Please check this typo!
In the title of Table 1 also need to add the term „and newborns with DS“ because this data is in the table
Author Response
Dear Reviewer,
We are very grateful for your reevaluation of our article “Why is health care for children with Down syndrome so crucial from the first days of life? A retrospective cohort study emphasized transient abnormal myelopoiesis (TAM) syndrome at three centers. We would like to respond to the comments and suggestions made.
Introduction
The authors in your response noticed that they moved the paragraph „Therefore, in this study we decided to analyze the frequency of medical problems, including hematological disorders, TAM syndrome and leukemia occurring in newborns with Down's syndrome born in three hospital centers during the 5 years. It was also checked whether children with DS were referred to a hematologist or oncologist.“ to the methods but this was not deleted and not moved. Please check this typo!
Thank you very much for your repeated comment – these sentences were moved to the beginning of Materials and Methods section. Sorry for this oversight.
In the title of Table 1 also need to add the term „and newborns with DS“ because this data is in the table.
Thank you for this comment, the title presents now like that:
”Table 1. Number of all live births and newborns with DS (in parentheses) between 2016 and 2020 in Gynecological and Obstetrics University Hospital (UH), Provincial Hospital of Poznan (PH), and Franciszek Raszeja City Hospital (CH).”
Once again – very thant you for your reevaluation and comments.
Regards,
Danuta Januszkiewicz-Lewandowska